# Xanthine Oxidase Inhibition and Anti-LDL Oxidation by Prenylated Isoflavones from *Flemingia philippinensis* Root

**DOI:** 10.3390/molecules25133074

**Published:** 2020-07-06

**Authors:** Jeong Yoon Kim, Yan Wang, Zuo Peng Li, Aizhamal Baiseitova, Yeong Jun Ban, Ki Hun Park

**Affiliations:** 1Division of Applied Life Science (BK21 Plus), IALS, Gyeongsang National University, Jinju 52828, Korea; foryou6633@gmail.com (J.Y.K.); dameng@126.com (Z.P.L.); aizhabaiseitova@gmail.com (A.B.); banyoung972@naver.com (Y.J.B.); 2College of Food and Biological Engineering, Qiqihar University, Qiqihar 161006, China; pitwang05@163.com

**Keywords:** *Flemingia philippinensis*, prenylated isoflavones, xanthine oxidase inhibition, anti-LDL oxidation

## Abstract

Xanthine oxidase is a frontier enzyme to produce oxidants, which leads to inflammation in the blood. Prenylated isoflavones from *Flemingia philippinensis* were found to display potent inhibition against xanthine oxidase (XO). All isolates (**1**–**9**) inhibited XO enzyme with IC_50_ ranging 7.8~36.4 μM. The most active isoflavones (**2**–**5**, IC_50_ = 7.8~14.8 μM) have the structural feature of a catechol motif in B-ring. Inhibitory behaviors were disclosed as a mixed type I mode of inhibition with *K*_I_ < *K*_IS_. Binding affinities to XO enzyme were evaluated. Fluorescence quenching effects agreed with inhibitory potencies (IC_50_s). The compounds (**2**–**5**) also showed potent anti-LDL oxidation effects in the thiobarbituric acid-reactive substances (TBARS) assay, the lag time of conjugated diene formation, relative electrophoretic mobility (REM), and fragmentation of apoB-100 on copper-mediated LDL oxidation. The compound **4** protected LDL oxidation with 0.7 μM in TBARS assay, which was 40-fold more active than genistein (IC_50_ = 30.4 μM).

## 1. Introduction

Reactive oxygen species (ROS) have received important consideration due to their adjusted role in signaling during inflammation [1,2]. These oxidants produced commonly by NAD(P)H oxidase, nitric oxide (NO) synthase and xanthine oxidoreductase (XOR) as well as environmental factors such as stress, smoking, etc. [3,4]. In particular, the xanthine oxidase (XO, EC 1.17.3.2), one of two XOR interconvertible forms, is a crucial enzyme in the catabolism of purine, which oxidizes hypoxanthine into xanthine followed by xanthine transformation to uric acid [5,6]. Also, XO is a frontier enzyme to produce oxidants (ROS), such as superoxide anion (O_2_^•−^) and hydrogen peroxide, because of its ability to transfer electron to molecular oxygen [7,8]. Endothelium-associated XO or XOR activity contributes to oxidative stress in blood vessels and inflammatory consequences arising from hypercholesterolemia and atherosclerosis [3,9]. Thus, controlling of XO excessive expression by its inhibitors—allopurinol and febuxostat—as the most widely prescribed commercial drugs, is important for human health, in particular, in regulation of vascular function [10]. However, they are often limited due to their inverse side effects, so the discoveries of new XO inhibitors are increasing.

On the other hand, there is a low-density lipoprotein (LDL) that is easily oxidized to oxLDL by ROS [11]. The oxLDL causes a form of cell accumulation and endothelial cell damages in the arterial intima, which finally promotes atherosclerosis [12,13]. The oxLDL stimulates monocytes to express cell surface adhesion of molecules causing vascular oxidative damage [14]. Moreover, oxLDL and the level of its receptors have been certainly associated with various types of cancer [15]. A representative anti-LDL substance, probucol has been used as antihyperlipidemic drug [16]. Thus, XO inhibitors and anti-LDL substances can contribute to suppress the production of ROS and oxidative damage of intima [17]. In this research, we explored effective antioxidant substances blocking XO enzymes as well as LDL oxidation.

The merit of *Flemingia philippinensis* lies on its wide cultivation in the southern part of China for commercial purposes as nutraceutical and food ingredient. The roots part has been traditionally used for curing rheumatism arthropathy and for improving the density of bone [18]. A bright representative of a legume family, *F. philippinensis* produces secondary metabolites that mostly consist of prenylated isoflavones [19,20], but it also produces various polyphenol compounds, namely benzofurans [21], flavanones [22], and chromenones [23] in low quantities. The isolated metabolites or extracts reported have the anti-inflammatory [18], antiestrogenic [24], and immunosuppressive features [25]. The polyphenolic compounds also showed great potential for enzyme inhibitions to bacterial neuraminidase [20], protein tyrosine phosphatase 1B [26], and human neutrophil elastase [27]. Recently, antioxidative potentials were investigated up to the protection of DNA damage [28].

This work aims to investigate antioxidant potentials of prenylated isoflavones based on xanthine oxidase (XO) inhibition and anti-LDL oxidation. The chromatographic feature and relative abundance of prenylated isoflavones were illustrated by UPLC-ESI-TOF/MS. XO inhibition was characterized by using secondary plots of the Michaelis-Menten equation and fluorescence quenching experiments. Besides, the inhibition of LDL oxidation by all isolates were investigated by four different assay methods including relative electrophoretic mobility (REM).

## 2. Results and Discussion

### 2.1. Isolation of Isoflavones from F. philippinensis

*F. philippinensis* is a polyphenol-enriched plant. Since the plant belongs to the legume family, the major metabolites of the plant are derivatives of isoflavone. These isoflavones have not only biological functions as enzyme inhibitors, but also antitumor activities relating to ROS and angiogenesis [29,30]. In our study, most of the isoflavones were obtained by examining antioxidant power targeted against XO enzyme and LDL oxidation. As a result, nine isoflavones were isolated through repeated chromatographies according to the previous description [20].

The isolates (**1**–**9**) were elicited as genistein (**1**), auriculasin (**2**), 6,8-diprenylorobol (**3**), 5,7,3′,4′-tetrahydroxy-2′,5′-di(3-methylbut-2-enyl)isoflavone (**4**), flemiphilippinin A (**5**), 5,7,3′-trihydroxy-2′-(3-methylbut-2-enyl)-4′,5′-(3,3-dimethylpyrano)isoflavone (**6**), 8-γ,γ-dimethylallylwighteone (**7**), osajin (**8**), and flemingsin (**9**) as shown in Figure 1. Their structures were elucidated by their spectroscopic data and comparing to those of previous reports [20,27]. Among the isolated compounds the most antioxidant, compound **4** had a molecular formula of C_25_H_26_O_6_ as established by the [M + H]^+^ ion at *m*/*z* 423.1806 (Calcd 422.1729) in the HR-ESI-MS analysis. The position of two prenyl appendages were clearly confirmed by HMBC correlations between H-1″ (*δ*_H_ = 3.30) and C-2′ (*δ*_C_ = 127.68), H-1′″ (*δ*_H_ = 3.35) and C-5′ (*δ*_C_ = 144.46).

### 2.2. Inhibitory Effect of Isolated Isoflavones on Xanthine Oxidase (XO) Activity

XO is an important enzyme capable of producing oxidants like ROS [3]. The dysregulation of this enzyme may lead to oxidative stress accompanied by impaired vascular function and subsequent cardiovascular diseases [5,9]. The activity of XO was screened by procedure represented in the previous report, where stepwise oxidation of xanthine (λ_max_ = 270 nm) to uric acid (λ_max_ = 295 nm) was observed [31]. All isolates (**1**–**9**) inhibited XO with an IC_50_ range of 7.8~36.4 μM. Dose-dependence of inhibitions was observed on all isolates as shown in Figure 2a. The most potent inhibitor **3** (IC_50_ = 7.8 μM) showed a similar tendency with commercially available inhibitor, allopurinol (Table 1). The subtle changes in structures slightly affected the activity. The catechol motif in B ring was preferred over the corresponding product of C4-hydroxyl group; **3** (IC_50_ = 7.8 μM) vs. **7** (IC_50_ = 16.9 μM). The prenyl groups in resorcinol motif within A ring contributed to increasing the inhibitory potency: **1** (IC_50_ = 25.0 μM) vs. **7** (IC_50_ = 16.9 μM). The products (**5** and **6**) of oxidative cyclization of the alcohol group with a prenyl group tend to decrease activities to 14.8 μM and 20.4 μM, respectively. The reversibility of inhibitor to the enzyme was deduced with the plots derived from initial velocity versus enzyme amount in the different concentrations (0~12.5 μM) of compound **3** (Figure 2b). They showed a typical reversible type of inhibition where all of the straight lines passed through the origin. Other compounds were also reversible inhibitors to XO enzyme with a similar pattern of compound **3**. The XO inhibitory behavior of prenylated isoflavones (**2**–**9**) was estimated using double-reciprocal plots (Appendix A).

All estimated prenylated isoflavones were observed as mixed type inhibitors. As shown in Figure 3a, representative inhibitor **3** was elucidated as a typically mixed type inhibitor, because the common intercept in Lineweaver–Burk plots was on the left of the vertical axis and above the horizontal axis. It indicates that *K*_m_ was increased, whereas *V*_max_ decreased with the increasing concentration of **3**. The secondary re-plots of the slope and intercept showed that *K*_I_ (5.7 μM) was much less than *K*_IS_ (15.5 μM). It is reasonable to conclude thus, compound **3** allowed as mixed type I, when inhibitor prefers free enzyme than substrate-enzyme complex. The *K*_i_ value of compound **3** was estimated to be 5.5 μM by Dixon plots (Figure 3b). Meanwhile, genistein (**1**) was estimated as competitive inhibitor due to constant *V*_max_ and enhancing *K*_m_ by the increasing concentration of **1** (Figure 3c), agreed to the previous report [32]. Dixon plots (Figure 3d) of **1** determined the *K*_i_ value with 11.5 μM.

### 2.3. Binding Affinity Between Xanthine Oxidase and Isolated Isoflavones

The XO enzyme has many numbers of fluorescent residues consist of 11 tryptophans, 34 tyrosines, and 68 phenylalanines according to [33] and Appendix A. Thus, the intrinsic fluorescence of XO enzyme is influenced by a function of the ligand concentration [32,33,34]. Fluorescence quenching (FS) effects were measured in the condition (excitation 290 nm, emission 300~400 nm) where there was no significant emission from any of the other components except enzyme. Results shown in Figure 4 demonstrates that FS intensities were reduced along with the inhibitor concentrations. Such wise, FS reduction tendencies had very close relations with inhibitory potencies (IC_50_ values). In particular, the compound **3** (IC_50_ = 7.8 μM) showed more rapid FS reduction than compound **9** (IC_50_ = 36.4 μM) and **1** (IC_50_ = 25.0 μM). The binding affinity constant (*K*_SV_) which was calculated using the Stern–Volmer equation (equation (5)) could be ranked in the order of inhibitory potencies as follow: **3** (IC_50_ = 7.8, *K*_SV_ = 1.2530) > **5** (IC_50_ = 14.8, *K*_SV_ = 0.5531) > **1** (IC_50_ = 25.0, *K*_SV_ = 0.1845) > **9** (IC_50_ = 36.4 μM, *K*_SV_ = 0.0930 × 10^5^ L·mol^−1^) given in Appendix A and Appendix A. A high correlation (R^2^ = 0.9879) between binding affinities (*K*_SV_) and inhibitory potencies (IC_50_) was observed (Figure 4d).

### 2.4. Protection Effects of Isoflavones on LDL Against Oxidative Damage

The isolated isoflavones (**1**–**9**) were evaluated in vitro for their antioxidant potential on Cu^2+^ induced LDL oxidation by using four different methods. All methods were carried out in the condition of human LDL (500 μg/mL) in the presence of 10 μM Cu^2+^ as an oxidation initiator according to the general procedure [35,36,37,38]. Firstly, in the thiobarbituric acid reactive substance (TBARS), all tested compounds exhibited antioxidant activities with a range of IC_50_ (0.7~30.4 μM) as shown in Table 2. In particular, four prenylated isoflavones **2**–**5** showed a distinctive inhibition to LDL oxidation with IC_50_ values 2.4, 1.9, 0.7, and 2.7 μM, respectively, which are 10–40 fold potent than genistein **1** (IC_50_ = 30.4). The prenyl groups contributed to antioxidant potential as like genistein **1** (IC_50_ = 30.4) vs. 6,8-diprenyl genistein **7** (IC_50_ = 14.6 μM).

Secondly, all isoflavones (**1**–**9**) were applied to the conjugated diene formation detecting at 234 nm for 180 min, of which lag time indicated the resistance of LDL oxidation. The control LDL (500 μg/mL) incubated with 10 μM CuSO_4_ had a lag time of 80 min, whereas antioxidants extended the lag time is proportional to their potential (Figure 5a). The lag times in Table 2 were associated with the results from TBARS assay as follows: **6** (l52 min, IC_50_ = 10.9 μM) > **9** (128 min, 14.6 μM) > **8** (108 min, 24.6 μM) > probucol (83 min, 35.1 μM). The four representative antioxidants **2**–**5** clearly showed to extend the lag time from 80 min to 180 min at 10 μM concentrations. Dose-dependent of antioxidant concentration was demonstrated with the most active compound **4**, of which lag time was extended to 120 min (0.3 μM), 160 min (0.6 μM), and 180 min (1.25 μM) as shown Figure 5b. The prenylated groups were proved to attribute LDL oxidation from genistein **1** (80 min) vs. its prenyl derivative **7** (124 min) at 10 μM concentration (Figure 5b).

An oxidized LDL was also observed by the relative electrophoretic mobility (REM) assay method. The REM assay in Figure 6a was designed as follows: lane 1 (native LDL), lane 2 (oxidized LDL with 10 μM CuSO_4_), and lane 3–12 (native LDL and 10 μM compounds with 10 μM CuSO_4_), which were incubated for 16 h. The mobility of the LDL was clearly reduced by compounds **6**, **7**, and **9** that had 10~15 μM of IC_50_s in TBARS method. The most potent anti-LDL activities were observed on compounds **2**–**5**, which illustrated that REM results had a high correlation with TBARS (Figure 6a and Table 2). Figure 6b showed a dose-dependent activity of compound **4** at three different concentrations. Some selective compounds (**2**–**5**) were applied to ApoB-100, which is a major component of LDL and could be fragmented by ROS. The fragmentation of ApoB-100 was evaluated by the electrophoretic analysis on 4% polyacrylamide gel as shown in Figure 6c and Appendix A. The compounds **2**–**5** (lanes 4–7) protected the ApoB-100 fragmentation completely against copper-induced LDL oxidation at 2 μM concentrations. The above four different methods proved that all isoflavones had antioxidant potentials to LDL oxidation. In particular, prenylated isoflavones (**2**–**5**) showed a distinctive inhibition of LDL oxidation at low concentrations.

### 2.5. Determination of Prenylated Isoflavones in the Plant by UPLC-Q-TOF-MS

MeOH extract of *F. philippinensis* was analyzed under optimized UPLC-Q-TOF-MS to estimate natural abundance of prenylated isoflavones in the plant. MS analysis and conditional optimization are crucial for obtaining fragment ions of the target compound. Nine isoflavones were deduced from the molecular formula of each chromatographic peak, fragment ion peak, the retention time of authentic compound, and reference [24,39] to the database (Appendix A). The base peak intensity (BPI) positive ion chromatogram showed a complete chromatographic separation of major and minor peaks within 8 min as shown in Figure 7.

Peak 4 was identified as 5,7,3′,4′-tetra-hydroxy-2′,5′-di(3-methylbut-2-enyl)isoflavone (**4**) which yielded [M + H]^+^ ion at *m*/*z* 423.1806 and a typical fragment ion [(M + H)–C_4_H_8_(prenyl)–C_4_H_8_(prenyl)]^+^ at *m*/*z* 311.0556. Peak 3 was structural isomer of peak 4 and determined as 6,8-diprenylorobol (**3**) with the same fragmentation pattern, [M + H]^+^ ion at *m*/*z* 423.1806 and a fragment ion at *m*/*z* 311.0557. Peaks 4 (t_R_ = 4.5 min) and 3 (t_R_ = 5.2 min) were distinguished by comparison of retention times of their authentic samples.

Peaks 6 (4.8 min) and 2 (5.8 min) were also structural isomers of each other. Peak 6 was found out as 5,7,3′-trihydroxy-2′-(3-methylbut-2-enyl)-4′,5′-(3,3-dimethyl-pyrano)isoflavone (**6**) with an [M + H]^+^ ion at *m*/*z* 421.1655 and a fragment ion at *m*/*z* 365.1025. Peak 2 was identified as auriculasin (**2**) having an [M + H]^+^ ion at *m*/*z* 421.1655 and a fragment ion at *m*/*z* 365.1025.

Peak 9 had an [M + H]^+^ ion at *m*/*z* 437.1968 and a fragment ion [M + H–56(prenyl)]^+^ at *m*/*z* 381.1398 to be flemingsin (**9**). Peak 7 had an [M + H]^+^ ion at *m*/*z* 407.1853 and a fragment ion [M + H–56–56]^+^ at *m*/*z* 295.0604 to be 8-γ,γ-dimethylallylwighteone (**7**). Peak 8 was identified as osajin (**8**) due to an [M + H]^+^ ion at *m*/*z* 405.1706 and a fragment ion [M + H–56]^+^ at *m*/*z* 349.1081. Peak 5 was identified as flemiphilippinin A (**5**) due to an [M + H]^+^ ion at *m*/*z* 489.2283, and fragment ions [M + H–56]^+^ at *m*/*z* 433.1650 and [M + H–56–68]^+^ at *m*/*z* 365.1001. These isoflavones (**2**–**9**) corresponded to the antioxidative effect in extract and were determined as predominant and/or moderate metabolites in *F. philippinensis*.

## 3. Materials and Methods

### 3.1. Plant Materials and Chemicals

Root barks of F. philippinensis were gathered from a farm in Ninning, Guanxi province, China. The plant was confirmed with Voucher specimens (No. 530) by Dr. Yimmin Zhao, Gaunxi Botanical Garden, China. It was cleaned, dried, and stored at a cool temperature (<18 °C) in darkness. Methanol, chloroform, butanol, *n*-hexane, sodium hydroxide (NaOH), and copper sulfate (CuSO_4_) were purchased from Duksan Co. (Gyeonggi-do, South Korea). Xanthine oxidase from bovine milk (EC 1.17.3.2), xanthine, allopurinol, sodium pyrophosphate, dimethyl sulfoxide (DMSO), probucol, thiobarbituric acid, trichloroacetic acid, and other used organic solvent were from Sigma Aldrich Co., (St. Louis, MO, USA). Low-density lipoprotein (LDL) was taken from Invitrogen (Thermo Fisher Scientific, Carlsbad, California, USA). ODS-C18 and Triart prep C18-S were from YMC Co., (Kyoto, Japan). Agarose, sodium dodecyl sulfate (SDS), *N*,*N*,*N′*,*N′*-tetramethylethylenediamine (TEMED), coomassie brilliant blue R-250, ammonium persulfate (APS), and sample buffer were purchased from Bio-Rad (Hercules, CA, USA).

### 3.2. Instruments

The isoflavones (**1**–**9**) from *F. philippinensis* were isolated by using Forte/R 100 MPLC (YMC Co., Ltd., Kyoto, Japan). The identification of compound structures was made based on ^1^H, ^13^C, DEPT-90, 135, COSY, HMQC, and HMBC spectra (Bruker 500MHz NMR, Billerica, MA, USA). All the experiments measuring of absorption and fluorescence were performed on a multi-mode microplate reader SpectraMax M3 (Molecular Devices, San Jose, CA, USA). Measurement of REM and ApoB-100 fragmentations were conducted using the Mini-Electrophoresis system (TAESHINE Bioscience, Gyeonggi-do, South Korea) and SDS-PAGE (Bio-Rad, Hercules, CA, USA), respectively. High-resolution electrospray ionization mass spectra (HR-ESI-MS) were obtained from Acquity-LC and Vion-MS (Waters, Milford, CT, USA).

### 3.3. Extraction and Isolation of Isoflavones from F. philippinensis

The metabolites were isolated using a slight modification to the reported method [20]. Briefly, the dried root barks of *F. philippinensis* (0.5 kg) were extracted with MeOH (5 L × 3) to give a dark red extract (73 g, 14.6%). The extract was dissolved in water and fractionated by the increasing polarity of the solvent system from hexane, chloroform up to butanol. The chloroform fraction (2 g) was subjected to MPLC over ODS-C18 packed glass column (35 × 250 mm, 40 μm), eluted with the gradient solvent system from 10 to 100% MeOH to yield ten fractions (A–J). It was repeated 7 times to yield isoflavones rich fractions: C 1.3 g; D 1.6 g; E 0.9 g. The fraction C was eluted by the gradient solvent system (40–100% MeOH) in recycle HPLC with Triart Prep C18-S (250 × 4.6 mm, S-10 μm, 12 nm) to yield compounds **1** (9mg), **4** (24 mg), **6** (16 mg) and **3** (28 mg). The fraction D was rechromatographed using recycle HPLC over ODS-C18 column in recycling system and the fraction was eluted with 60% MeOH to give compounds **9** (15 mg), **7** (21 mg) and **2** (22 mg). The fraction E was chromatographed on reversed silica (250 g, 25 μm) by 50 % MeOH to yield compounds **8** (9 mg) and **5** (11 mg). The structural identification data of isolated compounds (**1**–**9**) was presented in the Appendix A (Appendix A). Spectroscopic data of selected compounds (**2**–**5**) were given below.

#### 3.3.1. Auriculasin (**2**)

UV (MeOH) λ_max_ (log ε) 272 (3.54) nm; HR-ESI-MS 420.1582 [M]^+^ (calcd for C_25_H_24_O_6_ 420.1573); ^1^H-NMR (500 MHz, CDCl_3_): *δ* 1.49 (H-5″ and 6″), 1.69 (H-4′′′), 1.82 (H-5′′′), 3.35 (H-1′′′), 5.24 (H-2′′′), 5.60 (H-2″), 6.70 (H-1″), 6.78 (H-6′), 6.99 (H-5′), 7.29 (H-2′), 7.86 (H-2), 12.93 (OH-5).

#### 3.3.2. 6,8-Diprenylorobol (**3**)

UV (MeOH) λ_max_ (log ε) 260 (4.21), 280 (3.38) nm; HR-ESI-MS 422.1733 [M]^+^ (calcd for C_25_H_26_O_6_ 422.1729); ^1^H-NMR (500 MHz, Acetone-*d*_6_): *δ* 1.77 (H-4″ and 4′′′), 1.85 (H-5″ and 5′′′), 3.47 (H-1″), 3.49 (H-1′′′), 5.25 (H-2″ and 2′′′), 6.75 (H-6′), 6.80 (H-5′), 6.97 (H-2′), 7.90 (H-2), 12.90 (OH-5).

#### 3.3.3. 5,7,3′,4′-Tetrahydroxy-2′,5′-di(3-methylbut-2-enyl)isoflavone (**4**)

UV (MeOH) λ_max_ (log ε) 260 (4.50), 276 (2.96) nm; HR-ESI-MS 422.1733 [M]^+^ (calcd for C_25_H_26_O_6_ 422.1729); ^1^H-NMR (500 MHz, Acetone-*d*_6_), *δ* 1.45 (H-4″), 1.54 (H-5″), 1.71 (H-4′′′ and 5′′′), 3.30 (H-1″), 3.36 (H-1′′′), 5.08 (H-2″), 5.34 (H-2′′′), 6.30 (H-6), 6.43 (H-8), 6.54 (H-6′), 7.89 (H-2), 13.01 (OH-5).

#### 3.3.4. Flemiphilippinin A (**5**)

UV (MeOH) λ_max_ (log ε) 270 (2.32) nm; HRESIMS 488.2210 [M]^+^ (calcd for C_30_H_32_O_6_ 488.2199); ^1^H-NMR (500 MHz, CDCl_3_): *δ* 1.31 (H-4′′′′ and 5′′′′), 1.45 (H-4″ and 5″), 1.60 (H-4′′′), 1.72 (H-5′′′), 3.27 (H-1′′′), 5.02 (H-3′′′′), 5.15 (H-2′′′), 5.88 (H-2′′′′), 6.65 (H-1″), 6.68 (H-6′), 6.75 (H-5′), 6.90 (H-2′), 7.81 (H-2), 12.82 (OH-5).

### 3.4. Measurements of Xanthine Oxidase Activity

Bovine milk xanthine oxidase (XO, EC 1.17.3.2) activity was measured following the previous procedures [34,40]. In this assay, the formation of uric acid was measured spectrophotometrically at 295 nm at 37 °C, as a result of the reaction between xanthine oxidase and xanthine as a substrate. For the enzyme assay protocol, the tested isoflavones (**1**–**9**) and allopurinol as a positive control were dissolved in dimethyl sulfoxide (DMSO), and further diluted in different concentrations, from 0 to 2 mM, diluting twice at each step. The reaction mixture containing 160 μL of 200 mM sodium pyrophosphate/HCl (pH 7.5), 10 μL of diluted isoflavone or allopurinol, 20 μL of 300 μM xanthine (*K*_m_ = 30 μM), and 10 μL of 0.2 unit/mL bovine milk xanthine oxidase (final concentration: 0.01 unit/mL) was placed in clear bottom 96-well microplates and screened for 30 min every 30 s. The half-maximal inhibitory concentrations (IC_50_) were calculated according to the enzyme inhibition rate (%) derived from the change of uric acid formation in comparison with the concentration of inhibitors. The inhibition rate derived from the non-linear regression fitting to Equation (1).
Inhibition rate (%) = 100 × [(rate of control reaction − rate of inhibitor reaction)/rate of control reaction](1)

### 3.5. Xanthine Oxidase Inhibition Kinetics

The inhibition mode and inhibitory constants were determined by Lineweaver–Burk and Dixon plots. For obtaining double reciprocal plots, was measured the reaction absorbance of the enzyme, inhibited by different concentrations of inhibitors, with three concentrations of xanthine (15, 30, and 60 μM) at 295 nm and 37 °C. The corresponding kinetic parameters, namely the Michaelis-Menten constant (*K*_m_), maximal velocity (*V*_max_), inhibitory constant (*K*_i_), and dissociation constant of a binding site for the inhibitor to either free enzyme or the enzyme-substrate complex (*K*_I_ and *K*_IS_) were determined by Equations (2)–(4) [40].
1/*V* = *K*_m_/*V*_max_ (1 + [I]/*K*_I_) 1/S + 1/*V*_max_(2)
Slope = *K*_m_/*K*_I_*V*_max_ [I] + *K*_m_/*V*_max_(3)
Intercept = 1/*K*_IS_*V*_max_ [I] + 1/*V*_max_(4)

### 3.6. Fluorescence Quenching Measurements

The binding affinity between xanthine oxidase and isolated isoflavones were measured by fluorescence quenching method [33]. For the fluorescence quenching experiments, the Trp intensities of xanthine oxidase were monitored wavelength of emission from 300 to 400 nm and excitation of 290 nm. The reaction mixture containing 180 μL of 200 mM sodium pyrophosphate/HCl (pH 7.5), 10 μL of isolated isoflavones diluted in DMSO (3.125, 6.25, 12.5, 25.0, and 50.0 μM), and 10 μL of 1 unit/mL xanthine oxidase was placed in black/flat bottom 96-well microplates. The fluorescence emission spectra were observed in the absence (*F*_0_) or presence (*F*) of isoflavones as quenchers. The quenching parameters such as the Stern–Volmer constant (*K*_SV_), binding constant (*K*_A_), and the number of binding sites (*n*) were determined by the Equations (5) and (6) [41].
*F*_0_ − *F* = 1 + *K*_SV_ [*Q*](5)
Log [(*F*_0_ − *F*)/*F*] = log *K*_A_ + *n* log [*Q*]_f_(6)

### 3.7. Measurements of Thiobarbituric Acid Reactive Substances (TBARS)

Thiobarbituric acid reactive substances (TBARS) assay was carried out according to the previous method with minor modifications [38]. The absorbance of a product of a reaction between thiobarbituric acid (TBA) and malondialdehyde (MDA) was measured at 540 nm. First, 220 μL of 500 μg/mL LDL in a buffer (10 mM PBS, pH 7.4), 10 μL of 10 μM of CuSO_4_, and 10 μL of different concentrations from 0 to 3 mM of isoflavones or probucol as a positive control were mixed in 1.5 mL Eppendorf tube and incubated at 37 °C for 4 h. After incubation, 100 μL of trichloroacetic acid (TCA) and 100 μL of 0.67% thiobarbituric acid (TBA)/0.05 N NaOH were added to the reaction mixture and heated in the water bath at 100 °C for 15 min. After heating, the mixture was cooled on the ice and centrifuged for 15 min at 3000 rpm. The absorbance of supernatant was measured and the IC_50_ values were calculated from the inhibition rate derived from Equation (1).

### 3.8. Measurements of Formed Conjugated Dienes (CD)

The formation of conjugated dienes from oxLDL was assessed by the method [35]. The conjugated diene formation occurred in the clear bottom 96-well plate. The mixture of each well containing 230 μL of 550 μg/mL LDL in the buffer (10 mM PBS, pH 7.4) and 10 μL of 250 μM CuSO_4_ (final concentration: 10 μM) in the presence or absence of isoflavones or probucol as a positive control (final concentration: 10 μM) were incubated at 37 °C for 4 h. Then, the reaction was screened at 234 nm for 180 min every 5 min, continuously, incubating in the SpectraMax M3 multi-mode microplate reader. In the observed absorbance spectra, the lag time of tested samples was obtained as the endpoint indicated in the lag phase.

### 3.9. Measurements of Relative Electrophoretic Mobility (REM) and ApoB Fragmentation

The relative electrophoretic mobility (REM) of native LDL and oxidized LDL induced by Cu^2+^ was performed according to the procedures of references [35,36]. Before the electrophoresis, the samples containing 10 μM of CuSO_4_ and 10 μM of nine isoflavones or probucol were treated into LDL (500 μg/mL), then incubated for 16 h at 37 °C in dark condition. The native LDL and pretreated LDL samples loaded onto 0.5% agarose gel in TAE buffer (40 mM Tris-acetate with 1 mM EDTA, pH 8.0) and electrophoresed for 50 min at 100 V. After the electrophoresis, the gel was fixed in 40% ethanol with 10% acetic acid for 30 min, stained with 0.15% coomassie brilliant blue R250, and LDL bands visualized by using the destaining solution (methanol:acetic acid:water, 3:2:15, *v*/*v*).

The anti-LDL oxidation activity of four isoflavones (**2**–**5**) having the highest ratio of migrating distance from oxidized LDL in REM experiments was confirmed by measuring up the fragmentation of ApoB-100, LDL particle, according to the procedure [37]. The ApoB-100 fragmentation was performed on 4% SDS-PAGE for 150 min at 60 V. For the fragmentation assay sample contained native LDL, copper ion oxidized LDL in the presence or absence four isoflavones (final concentration: 2 μM) mixed with sample buffer, consisted of 3% SDS, 10% glycerol, and 5% 2-mercaptoethanol. The pretreated samples were heated for 5 min at 95 °C. The heated samples were cooled on ice and loaded on the SDS-PAGE in the buffer. After electrophoresis, the gel was stained and destained by using the same procedure with REM.

### 3.10. UPLC-ESI-TOF-MS Analysis

Metabolites of the *F. philippinensis* root barks methanol extract were analyzed by using ultra performance liquid chromatography quadrupole time-of-flight mass spectrometry (UPLC-Q-TOF-MS) equipped with sample manager-FTN, binary solvent manager from Acquity UPLC, and high-resolution ion mobility mass detector from Vion IMS Q Tof. On the base peak intensity (BPI) chromatogram, peaks of metabolites were separated by performing on Acquity UPLC BEH C_18_ column (2.1 mm × 100 mm, 1.7 μm; Waters) at 25 °C with gradient solvent system constituting the mobile phase A (0.1% formic acid in water) and B (acetonitrile). The linear gradient solvent system was as follows: 0–3 min, 50% B; 3–5 min, 70% B; 5–7 min, 90% B; 7–8 min, 100% B. The flow rate was 0.35 mL/min and the injection volume was 1 μL. The Q-TOF-MS was conducted in positive mode directly by passing to the ESI interface without splitting. The operating condition for mass analysis was 3 kV of capillary voltage, 40 V of sample cone voltage, 800 L/h of desolvation gas (Ar) flow, 30 L/h of cone gas (Ar) flow, desolvation temperature at 440 °C, and ion source temperature at 100 °C. The mass data collected by applying the energy (20–45 eV) was observed in the range from the mass-to-charge ratio 50 to 1500, with Leucine-enkephalin used as lock mass (566.2771 Da for positive mode) at the rate of 0.2 s/cycle. The assignment for the precursor and fragment ion patterns of the individual isoflavones (**1**–**9**) possessed by the methanol extract of *F. philippinensis* were assessed by the UniFi software (version 1.8.2.169, Waters) linked to databases (Chemspider, Metlin, and human metabolome).

### 3.11. Statistical Analysis

The triple repetition of all experiments proves the reliability of the results, which further verified in Sigma Plot (version 10.0, Systat Software, Inc., San Jose, CA, USA). Deviations from the values were considered significant at *p* < 0.05.

## 4. Conclusions

Xanthine oxidase (XO) is a key enzyme for producing oxidants (ROS), excessive amounts of which cause damage to cell structure such as like LDL oxidation. The prenylated isoflavones, major metabolites of *F. philippinensis*, were found to inhibit LDL oxidation as well as XO enzyme. The isoflavones had IC_50_ of 7.8~36.4 µM against XO enzymes with mixed type I inhibitory behavior. We also proved that the inhibitory potency rose from the binding affinity to the enzyme by fluorescence quenching effect. The most potent XO inhibitors (**2**–**5**) were proven to have potent anti-LDL activities by four different methods including REM assay. A natural abundance of antioxidative isoflavones (**2**–**9**) was demonstrated by BPI chromatogram of UPLC-ESI-TOF-MS. Disclosed antioxidant potentials would contribute to increasing the value of the target plant as a functional food ingredient.

## Figures and Tables

**Figure 1 molecules-25-03074-f001:**
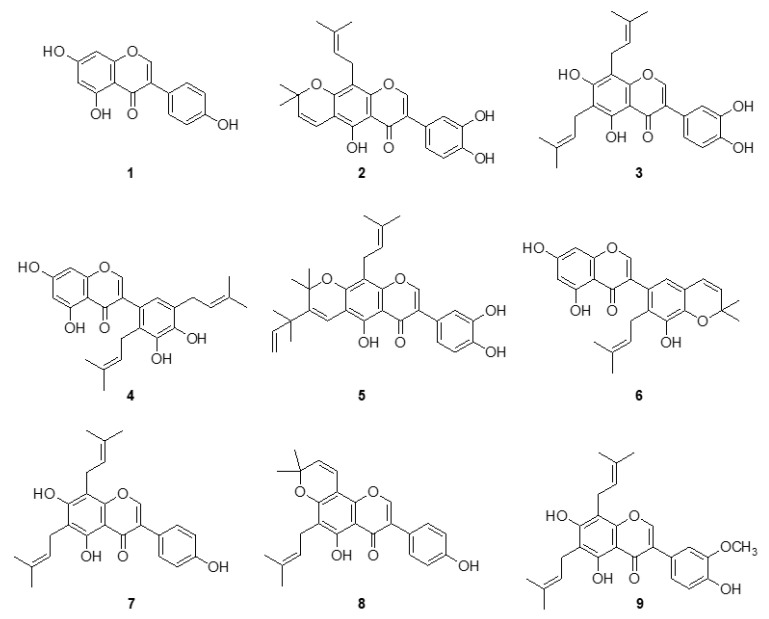
Chemical structures of the isolated isoflavones **1**–**9** from the root barks of *F. philippinensis*.

**Figure 2 molecules-25-03074-f002:**
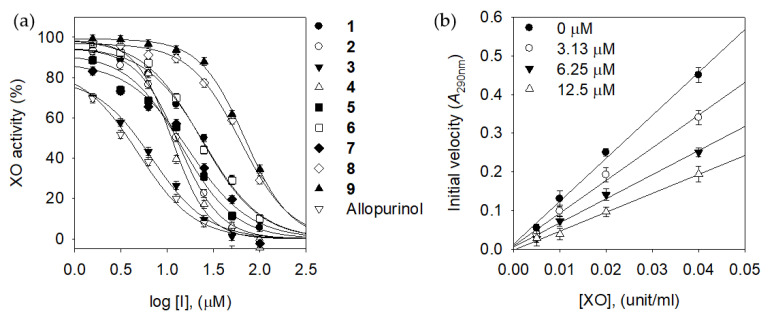
(**a**) Dose-dependent effects of the isolated compounds **1**–**9** and allopurinol on xanthine oxidase inhibition. (**b**) Determination of the reversible inhibitory mode of compound **3**.

**Figure 3 molecules-25-03074-f003:**
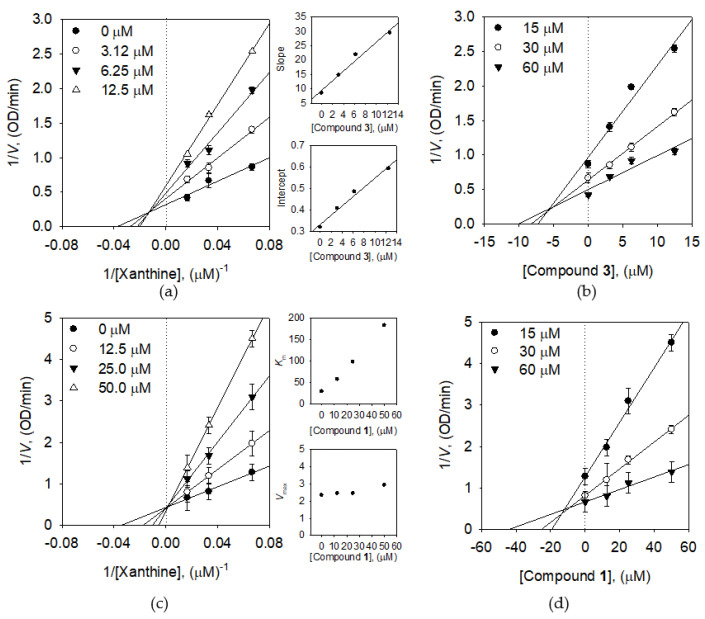
Enzymatic kinetics of compounds **3** and **1** using xanthine as a substrate. (**a**) Lineweaver–Burk plots of compound **3**. (Inset) The secondary plots of the straight lines of slope and intercept versus the concentration of compound **3** throughout Equations (2)–(4) are shown. (**b**) Dixon plots of compound **3**. (**c**) Lineweaver–Burk plots of compound **1** (Inset) Each curve provided tendencies for the Michaelis-Menten constant (*K*_m_) and maximal velocity (*V*_max_). (**d**) Dixon plots of compound **1**.

**Figure 4 molecules-25-03074-f004:**
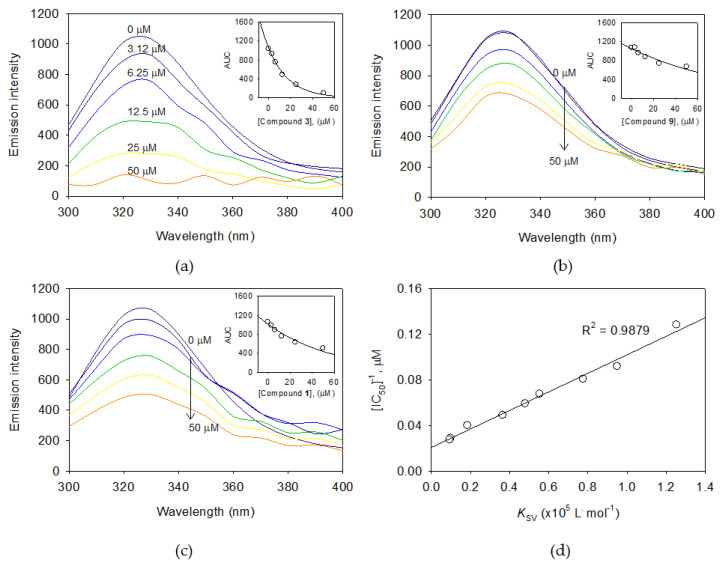
The fluorescence emission spectra of xanthine oxidase (XO) at different concentrations (0, 3.12, 6.25, 12.5, 25.0, and 50.0 μM) of (**a**) compound **3**, (**b**) compound **9**, and (**c**) compound **1**. (Inset) Normalized intensities of fluorescence for XO are shown for compounds **3**, **9**, and **1**, respectively. (**d**) The correlation between half-maximal inhibitory concentrations (IC_50_) values and Stern–Volmer constants (*K*_SV_) of compounds **1**–**9** is shown.

**Figure 5 molecules-25-03074-f005:**
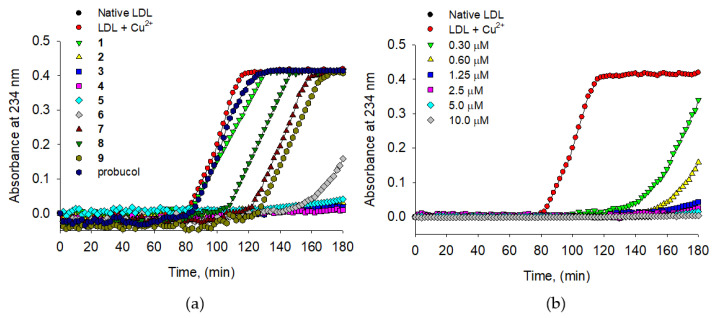
(**a**) Effects of isoflavones (**1**–**9**) on the generation of conjugated diene on Cu^2+^ induced LDL oxidation. (**b**) Dose-dependent effect of compound **4** on the generation of conjugated diene on Cu^2+^ induced LDL oxidation.

**Figure 6 molecules-25-03074-f006:**
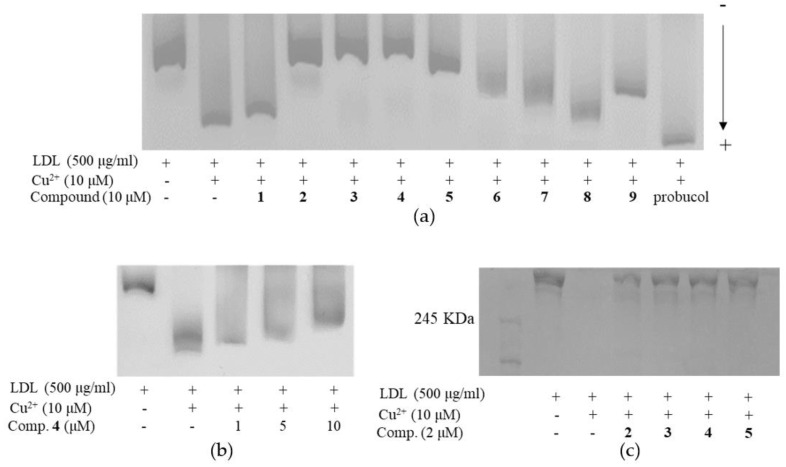
(**a**) Effect of isoflavones (**1**–**9**) on the Cu^2+^ mediated LDL oxidation by relative electrophoretic mobility (REM). (**b**) Dose-dependent effect of compound **4** against Cu^2+^ mediated LDL oxidation by REM. (**c**) Effects of isoflavones (**2**–**5**) on the ApoB-100 fragmentation.

**Figure 7 molecules-25-03074-f007:**
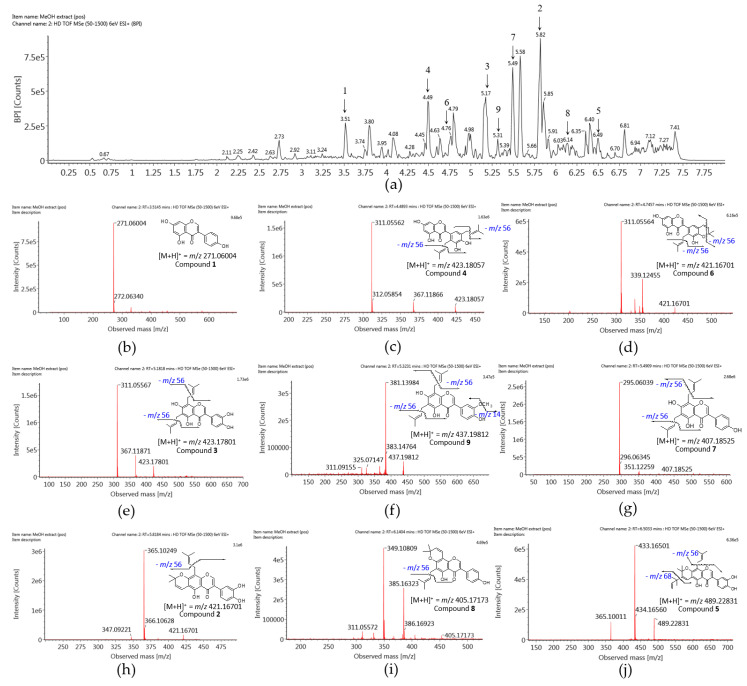
(**a**) The UPLC-ESI-TOF/MS chromatogram by base peak intensity (BPI) from the root barks of *F. philippinensis* methanol extract. Mass spectrum of nine isoflavones; (**b**) genistein (**1**); (**c**) 5,7,3′,4′-tetra-hydroxy-2′,5′-di(3-methylbut-2-enyl)isoflavone (**4**); (**d**) 5,7,3′-trihydroxy-2′-(3-methylbut-2-enyl)-4′,5′-(3,3-dimethylpyrano)isoflavone (**6**); (**e**) 6,8-diprenylorobol (**3**); (**f**) flemingsin (**9**); (**g**) 8-γ,γ-dimethyl-allylwighteone (**7**); (**h**) auriculasin (**2**); (**i**) osajin (**8**); (**j**) flemiphilippinin A (**5**).

**Table 1 molecules-25-03074-t001:** Xanthine oxidase inhibitory effects of compounds **1**–**9** isolated from *F. philippinensis*.

Compounds	IC_50_ ^a^ (µM)	Inhibition Mode (*K*_i_ ^b^, µM)	*K*_I_ (µM)	*K*_IS_ (µM)
**1** **2** **3** **4** **5** **6** **7** **8** **9** Allopurinol ^d^	25.0 ± 2.3 12.4 ± 0.8 7.8 ± 0.6 10.9 ± 0.4 14.8 ± 1.2 20.4 ± 0.2 16.9 ± 0.4 34.7 ± 1.6 36.4 ± 0.5 5.2 ± 0.1	Competitive (11.5 ± 0.4) Mixed type I (11.1 ± 0.5) Mixed type I (5.5 ± 0.3) Mixed type I (11.5 ± 0.3) Mixed type I (12.8 ± 0.2) Mixed type I (19.3 ± 0.8) Mixed type I (11.3 ± 0.3) Mixed type I (33.9 ± 1.3) Mixed type I (35.1 ± 0.9) NT	NT ^c^ 10.8 5.7 10.7 12.5 18.8 7.2 35.1 35.2 NT	NT 28.1 15.5 23.0 26.3 40.0 12.9 52.4 49.0 NT

All compounds were examined in a set of experiments repeated three times; ^a^ Sample concentration which leads to 50% xanthine oxidase activity loss; ^b^ Values of inhibition constant; ^c^ NT is not tested; ^d^ Allopurinol was used as a positive control.

**Table 2 molecules-25-03074-t002:** Effects of isoflavones (**1**–**9**) from *F. philippinensis* on oxidized LDL induced by Cu^2+^.

Compounds	IC_50_ ^a^ (μM)	Inhibition Rate ^b^ (%)	Lag Time ^c^ (min)
**1** **2** **3** **4** **5** **6** **7** **8** **9** Probucol ^d^	30.4 ± 1.5 2.4 ± 0.1 1.9 ± 0.3 0.7 ± 0.05 2.8 ± 0.5 10.9 ± 1.3 15.2 ± 0.3 24.6 ± 0.7 14.6 ± 0.6 35.1 ± 0.8	20.3 95.9 98.6 100 94.1 48.8 38.7 25.4 40.3 16.4	80 >180 >180 >180 >180 152 124 108 128 83

All compounds were examined in triplicate. ^a^ IC_50_ values of isoflavones represent the concentration that caused 50% oxidation LDL by measurement of TBARS assay; ^b^ Inhibition rate was expressed at 10 μM of isoflavones (**1**–**9**) concentration. ^c^ Lag time of conjugated diene formation by oxLDL at 10 μM of isoflavones (**1**–**9**) concentration. ^d^ Probucol was used as a positive control.

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
