# Peer review of "Xanthine Oxidase Inhibition and Anti-LDL Oxidation by Prenylated Isoflavones from Flemingia philippinensis Root"

_molecules, 2020, doi:10.3390/molecules25133074_

Round 1

Reviewer 1 Report

This manuscript (MS), molecules-851489, “Xanthine oxidase inhibition and anti-LDL oxidation by prenylated isoflavones from Flemingia philippinensis root” is an interesting observation that found the LDL oxidation inhibitory effect of prenylated isoflavones. The authors have found nine prenylated isoflavones from Flemingia philipponensis root, and 5,7,3',4'-tetrahydroxy-2',5'-di(3-methylbut-2-menyl) isoflavone exerts the strongest XO inhibitory effect among them. Furthermore, the authors have found that the catechol motif in B ring contributes to inhibit the activity of XO. Since the oxidation of LDL by XO is involved in the development of cardiovascular diseases, this authors’ finding is a very important result in maintaining human health, in particular regulation of vascular function. The MS is of good quality, and I have a comment for this MS. I encourage the authors to read the comments below and brush up the authors’ MS. I hope my comments will help the authors.

Comment.

I feel that Figure 6 is difficult to understand for readers. Because Lane No. and compound No.  are mixed in Figure 6. Each lane’s description is noted in the caption, but I think it’s easier for the reader to understand them if they are visually clear.

Author Response

Response to Reviewer 1

Q1. I feel that Figure 6 is difficult to understand for readers. Because Lane No. and compound No.  are mixed in Figure 6. Each lane’s description is noted in the caption, but I think it’s easier for the reader to understand them if they are visually clear.

→ Yes, Figure 6 and its captions were resharphened to be visually clear and understand easily for readers. Please see Figure 6 and its captions.

Reviewer 2 Report

In the manuscript “Xanthine Oxidase Inhibition and Anti-LDL Oxidation by Prenylated Isoflavones from Flemingia philippinensis root”, the authors investigate the bioactivity of nine isolated isoflavones from F. philippinensis root barks against xanthine oxidase and other in vitro oxidative assays.

In general, the manuscript is well-written and adequately describe the various experiments to investigate the anti-oxidative potential of prenylated isoflavones from the root bark of F. philippinensis, a well-known food ingredient with antioxidant properties.

Still, some comments should be considered before the work is accepted for publication:

  1. The number of the isolated isoflavones does not correspond to the peak name in table S2 (for example, auriculasin is called compound 2 in Figure 1 and is identified as peak 7 on table S2.)
  2. The purpose of the UPLC-Q-TOF-MS analysis was to estimate the natural abundance of prenylated isoflavones in the plant. However, there`s no estimative, that is, the relative quantitation of the peaks identified as the isolated isoflavones. In addition, several LC peaks have not been identified. It would be interesting to add the data of these peaks (m/z, fragments). Certainly more prenylated isoflavones could be tentatively annotated.
  3. The heterolytic cleavages are not correctly represented in figure 7, for example in (c), the neutral losses of two C4H8 can occur in both prenyl side-chains, which can lead to m/z 367 and m/z 311 both in C-3’ and C-6’ of ring B. Since the objective of this work is not exploring the gas-phase fragmentation reactions and the possible preference for one of the substituents in this competitive dissociative pathway, I`d encourage to adjust the figures to avoid misinterpretation and also to add references to previous studies focusing on characterization of prenylated flavonoids by MS/MS (DOI: 10.1002/rcm.6361 and others)
  4. The authors should improve quality resolution of Figure 7.
  5. I suggest the authors to make the MS/MS spectra of pure compounds in online databases such as GNPS (https://gnps.ucsd.edu/ProteoSAFe/static/gnps-splash.jsp). These data will allow the detection of these substances in other studies.

Author Response

Response to Reviewer 2

Q1. The number of the isolated isoflavones does not correspond to the peak name in table S2 (for example, auriculasin is called compound 2 in Figure 1 and is identified as peak 7 on table S2.)

→ The peak numbers were rearranged with order of compounds number, respectively. Please see Figure 7 and Table S2.

Q2. The purpose of the UPLC-Q-TOF-MS analysis was to estimate the natural abundance of prenylated isoflavones in the plant. However, there`s no estimative, that is, the relative quantitation of the peaks identified as the isolated isoflavones. In addition, several LC peaks have not been identified. It would be interesting to add the data of these peaks (m/z, fragments). Certainly more prenylated isoflavones could be tentatively annotated.

→ The relative quantification or abundance in mass chromatogram can only be obtained by peak area. We believe that BPI gram in Figure 7 has good enough retention times separating individual metabolite and peak area to see a relative abundance.

We annotated six more compounds in BPI gram which are not isoflavones. In here, we have focused on alkylated isoflavones. We can provide these annotations to the supplementary section if necessary. However, these data might make some confuse for reading.

Q3. The heterolytic cleavages are not correctly represented in figure 7, for example in (c), the neutral losses of two C4H8 can occur in both prenyl side-chains, which can lead to m/z 367 and m/z 311 both in C-3’ and C-6’ of ring B. Since the objective of this work is not exploring the gas-phase fragmentation reactions and the possible preference for one of the substituents in this competitive dissociative pathway, I`d encourage to adjust the figures to avoid misinterpretation and also to add references to previous studies focusing on characterization of prenylated flavonoids by MS/MS (DOI: 10.1002/rcm.6361 and others)

→ Yes, we agree with the reviewer’s suggestion completely. It is difficult to find which fragment corresponds to have first occurred. The signs of the fragmentation pattern of compounds (3, 4, 7, and 9) in Figure 7 were modified to avoid an incorrect interpretation. Furthermore, we added references to characterize target isoflavones more clearly. Please see Line 320 and Figure 7.

Q4. The authors should improve quality resolution of Figure 7.

→ Yes, the resolution quality was improved to be 500 DPI from 150 DPI. Please see Figure 7.

Q5. I suggest the authors to make the MS/MS spectra of pure compounds in online databases such as GNPS (https://gnps.ucsd.edu/ProteoSAFe/static/gnps-splash.jsp). These data will allow the detection of these substances in other studies.

→ Yes, we will try to upload eight target isoflavones (2-9) to databases such as GNPS.

Reviewer 3 Report

In this work authors analized the effects of the root of Flemingia philippinensis, a plant in the legume family Fabaceae. This research team have a widely experience in the study of Flemingia philippinensis during last seven years:

  • Kim, J. Y., Wang, Y., Uddin, Z., Song, Y. H., Li, Z. P., Jenis, J., & Park, K. H. (2018). Competitive neutrophil elastase inhibitory isoflavones from the roots of Flemingia philippinensis. Bioorganic chemistry, 78, 249–257. https://doi.org/10.1016/j.bioorg.2018.03.024
  • Kim, J. Y., Wang, Y., Song, Y. H., Uddin, Z., Li, Z. P., Ban, Y. J., & Park, K. H. (2018). Antioxidant Activities of Phenolic Metabolites from Flemingia philippinensis Merr. et Rolfe and Their Application to DNA Damage Protection. Molecules (Basel, Switzerland), 23(4), 816. https://doi.org/10.3390/molecules23040816
  • Wang, Y., Kim, J. Y., Song, Y. H., Li, Z. P., Yoon, S. H., Uddin, Z., Ban, Y. J., Lee, K. W., & Park, K. H. (2019). Highly potent bacterial neuraminidase inhibitors, chromenone derivatives from Flemingia philippinensis. International journal of biological macromolecules, 128, 149–157. https://doi.org/10.1016/j.ijbiomac.2019.01.105
  • Wang, Y., Curtis-Long, M. J., Lee, B. W., Yuk, H. J., Kim, D. W., Tan, X. F., & Park, K. H. (2014). Inhibition of tyrosinase activity by polyphenol compounds from Flemingia philippinensis roots. Bioorganic & medicinal chemistry, 22(3), 1115–1120. https://doi.org/10.1016/j.bmc.2013.12.047
  • Wang, Y., Curtis-Long, M. J., Yuk, H. J., Kim, D. W., Tan, X. F., & Park, K. H. (2013). Bacterial neuraminidase inhibitory effects of prenylated isoflavones from roots of Flemingia philippinensis. Bioorganic & medicinal chemistry, 21(21), 6398–6404. https://doi.org/10.1016/j.bmc.2013.08.049
  • Wang, Y., Yuk, H. J., Kim, J. Y., Kim, D. W., Song, Y. H., Tan, X. F., Curtis-Long, M. J., & Park, K. H. (2016). Novel chromenedione derivatives displaying inhibition of protein tyrosine phosphatase 1B (PTP1B) from Flemingia philippinensis. Bioorganic & medicinal chemistry letters, 26(2), 318–321. https://doi.org/10.1016/j.bmcl.2015.12.021
  • Cho, H. D., Lee, J. H., Moon, K. D., Park, K. H., Lee, M. K., & Seo, K. I. (2018). Auriculasin-induced ROS causes prostate cancer cell death via induction of apoptosis. Food and chemical toxicology : an international journal published for the British Industrial Biological Research Association, 111, 660–669. https://doi.org/10.1016/j.fct.2017.12.007
  • Cho, H. D., Moon, K. D., Park, K. H., Lee, Y. S., & Seo, K. I. (2018). Effects of auriculasin on vascular endothelial growth factor (VEGF)-induced angiogenesis via regulation of VEGF receptor 2 signaling pathways in vitro and in vivo. Food and chemical toxicology : an international journal published for the British Industrial Biological Research Association, 121, 612–621. https://doi.org/10.1016/j.fct.2018.09.025

This manuscript offer valuable information about the chemical interaction between isoflavonas, XOR enzyme and LDL particles, and will be interesting for molecules`s readers. However, some aspect could be improve in the Paper.

  1. Because Flemingia philippinensis compouses have demonstrated inhibitory effects on several enzymes, may be possible that this isoflavonas could have a negative effects on others metabolic pathways?
  2. Line 38-40: “Thus, excessive expression of XO should be controlled by its inhibitor for human health, in particular regulation of vascular function”. Could authors explain what is the inhibitors for XOR?
  3. Line 46: “anti-LDL substance”. Please, indicate this kind of substances.
  4. The references in the text should be revised. (Sun, Li, & Xu, 2017), (Li, Deng, Zhang, Shu, & Qin, 2011)
  5. Line 109-110: “The most potent inhibitor 3 (IC 50 = 7.8 μM)….”. Inhibitor 3 or 4?.
  6. Table2: Compound 4 showed a inhibitory rate >100%??
  7. Authors present many results but the discussion of these should be improve. May be possible reference some studies about the biological effects of Flemingia isoflavonas on culture cells or animals models?.

Author Response

Response to Reviewer 3

Q1. Because Flemingia philippinensis compouses have demonstrated inhibitory effects on several enzymes, may be possible that this isoflavonas could have a negative effects on others metabolic pathways?

→ Our laboratory has been focused on development of enzyme inhibitor. F. philippinensis displayed inhibition to different enzyme sources (bacterial neuraminidase, human neutrophil elastase, and protein tyrosine phosphatase 1B). It might be natural that some compounds happen to inhibit several different enzymes. We have not checked a negative effect with other bioassay systems.

Q2. Line 38-40: “Thus, excessive expression of XO should be controlled by its inhibitor for human health, in particular regulation of vascular function”. Could authors explain what is the inhibitors for XOR?

→ We added some sentences and reference to introduction part as follow: “Thus, controlling of XO excessive expression by its inhibitors, allopurinol and febuxostat as most widely prescribed commercial drugs application, is important for human health, in particular regulation of vascular function [10]. But they are often limited due to their inverse side effects, so the discoveries of new XO inhibitors are increasing.” Please see Lines 38-42.

Q3. Line 46: “anti-LDL substance”. Please, indicate this kind of substances.

→ We added sentence and reference to introduction part as follow: “A representative anti-LDL substance, probucol has been used as antihyperlipidemic drug [16].” Please see Lines 47-48.

Q4. The references in the text should be revised. (Sun, Li, & Xu, 2017), (Li, Deng, Zhang, Shu, & Qin, 2011)

→ We changed references (Sun, Li, & Xu, 2017) and (Li, Deng, Zhang, Shu, & Qin, 2011) to be the right format to Molecules. Please see Line 58 and 59.

Q5. Line 109-110: “The most potent inhibitor 3 (IC 50 = 7.8 μM)….”. Inhibitor 3 or 4?.

→ The compound 3 was most active to xanthine oxidase (Please see Table 1). While the compound 4 was most active one to anti-LDL oxidation (Please see Table 2).

Q6. Table2: Compound 4 showed a inhibitory rate >100%??

→ Yes, the description was inadequate to understand. We changed >100 to 100. Please see Table 2.

Q7. Authors present many results but the discussion of these should be improve. May be possible reference some studies about the biological effects of Flemingia isoflavonas on culture cells or animals models?.

→ Yes, compound 2 has a significant anti-tumor activity on cells and animal models. We described it on discussion part. Please see Line 72-74.
